# Diagnostic Performance of Artificial Intelligence-Based Computer-Aided Detection and Diagnosis in Pediatric Radiology: A Systematic Review

**DOI:** 10.3390/children10030525

**Published:** 2023-03-08

**Authors:** Curtise K. C. Ng

**Affiliations:** 1Curtin Medical School, Curtin University, GPO Box U1987, Perth, WA 6845, Australia; curtise.ng@curtin.edu.au or curtise_ng@yahoo.com.hk; Tel.: +61-8-9266-7314; Fax: +61-8-9266-2377; 2Curtin Health Innovation Research Institute (CHIRI), Faculty of Health Sciences, Curtin University, GPO Box U1987, Perth, WA 6845, Australia

**Keywords:** children, confusion matrix, convolutional neural network, deep learning, diagnostic accuracy, disease identification, image interpretation, machine learning, medical imaging, pneumonia

## Abstract

Artificial intelligence (AI)-based computer-aided detection and diagnosis (CAD) is an important research area in radiology. However, only two narrative reviews about general uses of AI in pediatric radiology and AI-based CAD in pediatric chest imaging have been published yet. The purpose of this systematic review is to investigate the AI-based CAD applications in pediatric radiology, their diagnostic performances and methods for their performance evaluation. A literature search with the use of electronic databases was conducted on 11 January 2023. Twenty-three articles that met the selection criteria were included. This review shows that the AI-based CAD could be applied in pediatric brain, respiratory, musculoskeletal, urologic and cardiac imaging, and especially for pneumonia detection. Most of the studies (93.3%, 14/15; 77.8%, 14/18; 73.3%, 11/15; 80.0%, 8/10; 66.6%, 2/3; 84.2%, 16/19; 80.0%, 8/10) reported model performances of at least 0.83 (area under receiver operating characteristic curve), 0.84 (sensitivity), 0.80 (specificity), 0.89 (positive predictive value), 0.63 (negative predictive value), 0.87 (accuracy), and 0.82 (F1 score), respectively. However, a range of methodological weaknesses (especially a lack of model external validation) are found in the included studies. In the future, more AI-based CAD studies in pediatric radiology with robust methodology should be conducted for convincing clinical centers to adopt CAD and realizing its benefits in a wider context.

## 1. Introduction

Artificial intelligence (AI) is an active research area in radiology [1,2,3,4]. However, the investigation of use of AI for computer-aided detection and diagnosis (CAD) in radiology started in 1955. Any CAD systems are AI applications and can be subdivided into two types: computer-aided detection (CADe) and computer-aided diagnosis (CADx) [5,6,7]. The former focuses on the automatic detection of anomalies (e.g., tumor, etc.) on medical images, while the latter is capable of automatically characterizing anomaly types such as benign and malignant [7]. Since the 1980s, more researchers have become interested in the CAD system development due to availabilities of digital medical imaging and powerful computers. The first CAD system approved by The United States of America Food and Drug Administration was commercially available in 1998 for breast cancer detection [6].

Early AI-based CAD systems in radiology were entirely rule based, and their algorithms could not improve automatically. In contrast, machine learning (ML)-based and deep learning (DL)-based CAD systems can automatically improve their performances through training, and hence, they have become dominant. DL is a subset of ML, and its models have more layers than those of ML. The DL algorithms are capable of modeling high-level abstractions in medical images without predetermined inputs [5,8,9].

A recent systematic review has shown that the DL-based CAD systems in radiology have been developed for a range of areas including breast, cardiovascular, gastrointestinal, hepatological, neurological, respiratory, rheumatic, thyroid and urologic diseases, and trauma. The performances of these CAD systems matched expert readers’ capabilities (pooled sensitivity and specificity: 87.0% vs. 86.4% and 92.5% vs. 90.5%), respectively [10]. Apparently, the current AI-based CAD systems might help to address radiologist shortage problems [9,10,11]. Nevertheless, various systematic reviews have criticized that the diagnostic performance figures reported in many AI-based CAD studies were not trustworthy because of their methodological weaknesses [10,12,13].

Pediatric radiology is a subset of radiology [14,15,16,17]. The aforementioned systematic review findings may not be applicable to the pediatric radiology [10,12,13,16,17]. For example, the AI-based CAD systems for breast and prostate cancer detections seem not relevant to children [10,12,13,17]. Although the AI-based CAD is an important topic area in radiology [10,12,13], apparently, only two narrative reviews about various uses of AI in pediatric radiology (e.g., examination booking, image acquisition and post-processing, CAD, etc.) [17] and AI-based CAD in pediatric chest imaging have been published to date [16]. Hence, it is timely to conduct a systematic review about the diagnostic performance of AI-based CAD in pediatric radiology. The purpose of this article is to systematically review the original studies to answer the question: “What are the AI-based CAD applications in pediatric radiology, their diagnostic performances and methods for their performance evaluation?”

## 2. Materials and Methods

This systematic review of the diagnostic performance of the AI-based CAD in pediatric radiology was conducted as per the preferred reporting items for systematic reviews and meta-analyses (PRISMA) guidelines and patient/population, intervention, comparison, and outcome model. This involved a literature search, article selection, and data extraction and synthesis [10,12,13,14,18].

### 2.1. Literature Search

The literature search with the use of electronic scholarly publication databases, including EBSCOhost/Cumulative Index of Nursing and Allied Health Literature Ultimate, Ovid/Embase, PubMed/Medline, ScienceDirect, Scopus, SpringerLink, Web of Science, and Wiley Online Library was conducted on 11 January 2023 to identify articles investigating the diagnostic performance of the AI-based CAD in the pediatric radiology with no publication year restriction [12,19,20]. The search statement used was (“Artificial Intelligence” OR “Machine Learning” OR “Deep Learning”) AND (“Computer-Aided Diagnosis” OR “Computer-Aided Detection”) AND (“Pediatric” OR “Children”) AND (“Radiology” OR “Medical Imaging”). The keywords used in the search were based on the review focus and systematic reviews on the diagnostic performance of the AI-based CAD in radiology [19,20,21,22,23].

### 2.2. Article Selection

A reviewer with more than 20 years of experience in conducting literature reviews was involved in the article selection process [14,24]. Only peer-reviewed original research articles that were written in English and focused on the AI-based CAD in pediatric radiology with the diagnostic accuracy measures were included. Gray literature, conference proceedings, editorials, review, perspective, opinion, commentary, and non-peer-reviewed (e.g., those published via the arXiv research-sharing platform, etc.) articles were excluded because this systematic review focused on the diagnostic performance of the AI-based CAD in the pediatric radiology and appraisal of the associated methodology reported in the refereed original articles. Papers mainly about image segmentation or clinical prediction instead of disease identification or classification were also excluded [12].

Figure 1 illustrates the details of the article selection process. A three-stage screening process through assessing (1) article titles, (2) abstracts, and (3) full texts against the selection criteria was employed after duplicate article removal from the results of the database search. Every non-duplicate article within the search results was retained until its exclusion could be decided [14,25,26].

### 2.3. Data Extraction and Synthesis

Two data extraction forms (Table 1 and Table 2) were developed based on a recent systematic review on the diagnostic performance of AI-based CAD in radiology [12]. The data, including author name and country, publication year, imaging modality, diagnosis, diagnostic performance of AI-based CAD system (area under receiver operating characteristic curve (AUC), sensitivity, specificity, positive predictive value (PPV), negative predictive value (NPV), accuracy and F1 score), AI type (such as ML and DL) and model (e.g., support vector machine, convolutional neural network (CNN), etc.) for developing the CAD system, study design (either prospective or retrospective), source (such as public dataset by Guangzhou Women and Children’s Medical Center, China) and size (e.g., 5858 images, etc.) of dataset for testing the CAD system, patient/population (such as 1–5-year-old children), any sample size calculation, model internal validation type (e.g., 10-fold cross-validation, etc.), any model external validation (i.e., any model testing with use of dataset not involved in internal validation and acquired from different setting), reference standard for ground truth establishment (such as histology and expert consensus), any model performance comparison with clinician and model commercial availability were extracted from each included paper. When diagnostic performance findings were reported for multiple AI-based CAD models in a study, only the values of the best performing model were presented [27]. Meta-analysis was not conducted because this systematic review covered a range of imaging modalities and pathologies, and hence, high study heterogeneity was expected, affecting its usefulness [12,13,28]. The Revised Quality Assessment of Diagnostic Accuracy Studies (QUADAS-2) tool was used to assess the quality of all included studies [9,12,13,19,23,27,29].

## 3. Results

Twenty-three articles met the selection criteria and were included in this review [30,31,32,33,34,35,36,37,38,39,40,41,42,43,44,45,46,47,48,49,50,51,52]. Table 1 shows their AI-based CAD application areas in the pediatric radiology and the diagnostic performances. These studies covered brain (*n* = 9) [30,31,32,33,34,35,36,37,38], respiratory (*n* = 9) [42,43,44,45,46,47,48,49,50], musculoskeletal (*n* = 2) [40,41], urologic (*n* = 2) [51,52] and cardiac imaging (*n* = 1) [39]. The commonest AI-based CAD application area (30.4%, 7/23) was pediatric pneumonia [43,45,46,47,48,49,50]. No study reported all seven diagnostic accuracy measures [30,31,32,33,34,35,36,37,38,39,40,41,42,43,44,45,46,47,48,49,50,51,52]. Most commonly, the papers (30.4%, 7/23) reported four metrics [30,32,35,42,44,45,52]. Accuracy (*n* = 19) and sensitivity (*n* = 18) were the two most frequently used evaluation metrics [30,31,32,33,34,35,36,37,38,39,41,42,43,44,45,46,47,48,49,50,51,52]. One study only used one measure, AUC [40]. Most of the articles (93.3%, 14/15; 77.8%, 14/18; 73.3%, 11/15; 80.0%, 8/10; 66.6%, 2/3; 84.2%, 16/19; 80.0%, 8/10) reported AI-based CAD model performances of at least 0.83 (AUC), 0.84 (sensitivity), 0.80 (specificity), 0.89 (PPV), 0.63 (NPV), 0.87 (accuracy), and 0.82 (F1 score), respectively. The ranges of the reported performance values were 0.698–0.999 (AUC), 0.420–0.987 (sensitivity), 0.585–1.000 (specificity), 0.600–1.000 (PPV), 0.260–0.971 (NPV), 0.643–0.986 (accuracy), and 0.626–0.983 (F1 score) [30,31,32,33,34,35,36,37,38,39,40,41,42,43,44,45,46,47,48,49,50,51,52]. For the seven studies about AI-based CAD for pneumonia, their model performances were at least 0.850 (AUC), 0.760 (sensitivity), 0.800 (specificity), 0.891 (PPV), 0.905 (accuracy) and 0.903 (F1 score).
children-10-00525-t001_Table 1Table 1Artificial intelligence-based computer-aided detection and diagnosis application areas in pediatric radiology and their diagnostic performances.Author, Year and CountryModalityDiagnosisDiagnostic PerformanceAUCSensitivitySpecificityPPVNPVAccuracyF1 ScoreBrain ImagingDou et al. (2022)—China [30]MRIBipolar disorder0.8300.9090.769NRNR0.854NRKuttala et al. (2022)—Australia, India & United Arab Emirates [31]MRIADHD and ASD0.850 (ADHA); 0.910 (ASD)NRNRNRNR0.854 (ADHA); 0.978 (ASD)NRLi et al. (2020)—China [32]MRIPosterior fossa tumors0.8650.9290.800NRNR0.878NRPeruzzo et al. (2016)—Italy [33]MRIMalformations of corpus callosum0.9530.9230.9040.906NR0.914NRPrince et al. (2020)—USA [34]CT & MRIACP0.978NRNRNRNR0.979NRTan et al. (2013)—USA [35]MRICongenital sensori-neural hearing loss0.9000.8900.860NRNR0.870NRXiao et al. (2019)—China [36]MRIASDNR0.9800.9360.9590.9710.963NRZahia et al. (2020)—Spain [37]MRIDyslexiaNR0.7500.7140.600NR0.7270.670Zhou et al. (2021)—China [38]MRIADHD0.6980.6090.676NRNR0.6430.626Cardiac ImagingLee et al. (2022)—South Korea [39]USKawasaki diseaseNR0.8410.5850.8110.6330.7590.826Musculoskeletal ImagingPetibon et al. (2021)—Canada, Israel and USA [40]SPECTLow back pain0.830NRNRNRNRNRNRSezer and Sezer (2020)—France and Turkey [41]USDDHNR0.9620.980NRNR0.977NRRespiratory ImagingBehzadi—Khormouji et al. (2020)—Iran and USA [42]X-rayPulmonary consolidation0.9950.9870.864NRNR0.945NRBodapati and Rohith (2022)—India [43]X-rayPneumonia0.939NRNRNRNR0.9480.959Helm et al. (2009)—Canada, UK and USA [44]CTPulmonary nodulesNR0.4201.0001.0000.260NRNRJiang and Chen (2022)-China [45]X-rayPneumoniaNR0.894NR0.918NR0.9120.903Liang and Zheng (2020)-China [46]X-rayPneumonia0.9530.967NR0.891NR0.9050.927Mahomed et al. (2020)-Netherlands and South Africa [47]X-rayPrimary-endpoint pneumonia0.8500.7600.800NRNRNRNRShouman et al. (2022)-Egypt and Saudi Arabia [48]X-rayBacterial and viral pneumonia0.9990.9870.9870.979NR0.9860.983Silva et al. (2022)-Brazil [49]X-rayPneumoniaNR0.945NR0.957NRNR0.951Vrbančič and Podgorelec (2022)-Slovenia [50]X-rayPneumonia0.9520.9760.9270.973NR0.9630.974Urologic ImagingGuan et al. (2022)-China [51]USHydronephrosisNRNRNRNRNR0.8910.895Zheng et al. (2019)-China and USA [52]USCAKUT0.9200.860.880NRNR0.870NRACP, adamantinomatous craniopharyngioma; ADHD, attention deficit hyperactivity disorder; ASD, autism spectrum disorder; AUC, area under receiver operating characteristic curve; CAKUT, congenital abnormalities of kidney and urinary tract; CT, computed tomography; DDH, developmental dysplasia of hip; MRI, magnetic resonance imaging; NPV, negative predictive value; NR, not reported; PPV, positive predictive value; SPECT, single-photon emission computed tomography; UK, United Kingdom; US, ultrasound; USA, United States of America.


Table 2 presents the included study characteristics. Overall, 18 out of 23 (78.3%) studies were published in the last three years [30,31,32,34,37,38,39,40,41,42,43,45,46,47,48,49,50,51]. Most of them (72.7%, 16/22) developed the DL-based CAD systems [31,34,36,37,39,40,41,42,43,45,46,48,49,50,51,52]. Of these 16 DL-based systems, 75% (*n* = 12) used the CNN model [34,37,39,40,41,42,43,46,48,49,50,51]. Magnetic resonance imaging (MRI) (*n* = 9) [30,31,32,33,34,35,36,37,38] and X-ray (*n* = 8) [42,43,45,46,47,48,49,50] were most frequently used by the AI-based CAD models for the brain and respiratory disease diagnoses, respectively. The majority of studies (69.6%, 16/23) collected the datasets retrospectively [31,33,34,36,38,39,40,42,43,44,45,46,48,49,50,52]. Of these 16 retrospective studies, about one-third (*n* = 11) relied on the public datasets [31,34,36,38,42,43,45,46,48,49,50]; most of them (*n* = 7) used the chest X-ray dataset consisting of 1741 normal and 4346 pneumonia images of 6087 1–5-year-old children collected from the Guangzhou Women and Children’s Medical Center, China [42,43,45,46,48,49,50]. No study calculated the sample size for the data collection [30,31,32,33,34,35,36,37,38,39,40,41,42,43,44,45,46,47,48,49,50,51,52]. Most of the studies (60.9%, 14/23) collected less than 233 cases [30,31,32,33,34,35,36,37,38,39,40,41,44,52], and about one-third (*n* = 7) collected data of less than 87 patients for testing their systems [30,32,34,35,37,40,44]. Hence, for the model internal validation, more than half of the studies (*n* = 13) used the cross-validation to address the small test set issue [30,33,34,35,36,37,38,39,40,47,50,51,52]. However, all but one did not conduct the external validation [30,31,32,33,34,35,36,37,38,39,40,41,42,43,45,46,47,48,49,50,51,52]. The only exception conducted external validation for a commercial AI-based CAD system evaluation [44]. Less than one-fifth of the included studies (*n* = 4) used the consensus diagnosis as the reference standard (ground truth) for the model training and performance evaluation [33,42,44,47], and one-quarter (*n* = 6) did not report the reference standard [31,43,45,46,48,49]. Only about one-fifth (*n* = 5) compared their model performances with those of clinicians [33,34,40,44,47], and most of these (60%, 3/5) were the studies using the consensus diagnosis as the reference standard [33,44,47].

Figure 2 shows the quality assessment summary of all (23) studies based on the QUADAS-2 tool. Only around one-third of the studies had a low risk of bias [34,35,36,37,38,41,44,52] and concern regarding applicability for the patient selection category [30,34,35,36,37,38,41,44,52]. The low risk of bias of the reference standard was only noted in about half of them [32,33,34,35,36,37,38,40,42,47,50,52].
children-10-00525-t002_Table 2Table 2Study characteristics of artificial intelligence-based computer-aided detection and diagnosis in pediatric radiology.Author, Year and CountryModalityDiagnosisAI Type and ModelStudy DesignDataset SourceTest Set SizePatient/PopulationSample Size CalculationInternal Validation TypeExternal ValidationReference StandardAI vs. ClinicianCommercial AvailabilityBrain ImagingDou et al. (2022)—China [30]MRIBipolar disorderML-LRProspectivePrivate dataset by Second Xiangya Hospital, China52 scans12–18-year-old childrenNo2-fold cross-validationNoClinical diagnosisNoNoKuttala et al. (2022)—Australia, India and United Arab Emirates [31]MRIADHD and ASDDL-GAN and softmaxRetrospectivePublic datasets (ADHD-200 and Autism Brain Imaging Data Exchange II)217 scansChildren (median ages for baseline and follow-up scans: 12 and 15 years, respectively)NoNRNoNRNoNoLi et al. (2020)—China [32]MRIPosterior fossa tumorsML-SVMProspectivePrivate dataset by Affiliated Hospital of Zhengzhou University, China45 scans0–14-year-old childrenNoRepeated hold-out with 70:30 random splitNoHistologyNoNoPeruzzo et al. (2016)—Italy [33]MRIMalformations of corpus callosumML-SVMRetrospectivePrivate dataset by Scientific Institute “Eugenio Medea”, Italy104 scans2–12-year-old childrenNoLeave-one-out cross validationNoExpert consensusYesNoPrince et al. (2020)—USA [34]CT and MRIACPDL-CNNRetrospectivePublic dataset (ATPC Consortium) and private datasets by Children’s Hospital Colorado and St. Jude Children’s Research Hospital, USA86 CT-MRI scansChildrenNo60:40 random split and 5-fold cross validationNoHistologyYesNoTan et al. (2013)—USA [35]MRICongenital sensori-neural hearing lossML-SVMProspectivePrivate dataset by Cincinnati Children’s Hospital Medical Center, USA39 scans8–24-month-old childrenNoLeave-one-out cross-validationNoFollow-upNoNoXiao et al. (2019)—China [36]MRIASDDL-SAE and softmaxRetrospectivePublic dataset (Autism Brain Imaging Data Exchange II)198 scans5–12-year-old childrenNo11-, 33-, 66-, 99- and 198-fold cross-validationNoClinical diagnosisNoNoZahia et al. (2020)—Spain [37]MRIDyslexiaDL-CNNProspectivePrivate dataset by University Hospital of Cruces, Spain55 scans9–12-year-old childrenNo4-fold cross validationNoClinical diagnosisNoNoZhou et al. (2021)—China [38]MRIADHDML-SVMRetrospectivePublic dataset (Adolescent Brain Cognitive Development Data Repository)232 scans9–10-year-old childrenNo10-fold cross-validationNoClinical diagnosisNoNoCardiac ImagingLee et al. (2022)—South Korea [39]USKawasaki diseaseDL-CNNRetrospectivePrivate dataset by Yonsei University Gangnam Severance Hospital, South Korea203 scansChildrenNo10-fold cross-validationNoSingle expert readerNoNoMusculoskeletal ImagingPetibon et al. (2021)—Canada, Israel and USA [40]SPECTLow back painDL-CNNRetrospectivePrivate dataset by Boston Children’s Hospital, USA65 scans10–17 years old childrenNo3-fold cross-validationNoOther-ground truth established by artificial lesion insertionYesNoSezer and Sezer (2020)—France and Turkey [41]USDDHDL-CNNProspectivePrivate dataset203 scans0–6-month-old childrenNo70:30 random splitNoSingle expert readerNoNoRespiratory ImagingBehzadi—Khormouji et al. (2020)—Iran and USA [42]X-rayPulmonary consolidationDL-CNNRetrospectivePublic dataset by Guangzhou Women and Children’s Medical Center, China582 images1–5-year-old childrenNo90:10 random splitNoExpert consensusNoNoBodapati and Rohith (2022)—India [43]X-rayPneumoniaDL-CNN and CapsNetRetrospectivePublic dataset by Guangzhou Women and Children’s Medical Center, China640 images1–5-year-old childrenNoNRNoNRNoNoHelm et al. (2009)—Canada, UK and USA [44]CTPulmonary nodulesNRRetrospectivePrivate dataset by a tertiary pediatric hospital29 scans3 years and 11 months to 18-year-old childrenNoNRYesExpert and reader consensusYesYesJiang and Chen (2022)—China [45]X-rayPneumoniaDL-ViTRetrospectivePublic dataset by Guangzhou Women and Children’s Medical Center, China624 images1–5-year-old childrenNoNRNoNRNoNoLiang and Zheng (2020)—China [46]X-rayPneumoniaDL-CNNRetrospectivePublic dataset by Guangzhou Women and Children’s Medical Center, China624 images1–5-year-old childrenNo90:10 random splitNoNRNoNoMahomed et al. (2020)—Netherlands and South Africa [47]X-rayPrimary-endpoint pneumoniaML-SVMProspectivePrivate dataset by Chris Hani Baragwanath Academic Hospital, South Africa858 digitized images1–59-month-old childrenNo10-fold cross-validationNoReader consensusYesNoShouman et al. (2022)—Egypt and Saudi Arabia [48]X-rayBacterial and viral pneumoniaDL-CNN and LSTMRetrospectivePublic dataset by Guangzhou Women and Children’s Medical Center, China586 images1–5-year-old childrenNo90:10 random splitNoNRNoNoSilva et al. (2022)—Brazil [49]X-rayPneumoniaDL-CNNRetrospectivePublic dataset by Guangzhou Women and Children’s Medical Center, China1172 images1–5-year-old childrenNoNRNoNRNoNoVrbančič and Podgorelec (2022)—Slovenia [50]X-rayPneumoniaDL-CNN and SGDRetrospectivePublic dataset by Guangzhou Women and Children’s Medical Center, China5858 images1–5-year-old childrenNo10-fold cross-validationNoExpert readersNoNoUrologic ImagingGuan et al. (2022)—China [51]USHydronephrosisDL-CNNProspectivePrivate dataset by Beijing Children’s Hospital, China3257 imagesChildrenNo10-fold cross-validationNoReaders and experts without consensusNoNoZheng et al. (2019)—China and USA [52]USCAKUTDL-SVMRetrospectivePrivate dataset by Children’s Hospital of Philadelphia, USA100 scansChildren with mean age of 111 days (SD: 262)No10-fold cross-validationNoClinical diagnosisNoNoACP, adamantinomatous craniopharyngioma; ADHD, attention deficit hyperactivity disorder; AI, artificial intelligence; ASD, autism spectrum disorder; ATPC, Advancing Treatment for Pediatric Craniopharyngioma; CAKUT, congenital abnormalities of kidney and urinary tract; CapsNet, capsule network; CNN, convolutional neural network; CT, computed tomography; DDH, developmental dysplasia of hip; DL, deep learning; GAN, generative adversarial network; LR, logistic regression; LSTM, long short-term memory; ML, machine learning; MRI, magnetic resonance imaging; NR, not reported; SAE, stacked auto-encoder; SD, standard deviation; SGD, stochastic gradient descent; SPECT, single-photon emission computed tomography; SVM, support vector machine; UK, United Kingdom; US, ultrasound; USA, United States of America; ViT, vision transformer.


## 4. Discussion

This article is the first systematic review on the diagnostic performance of the AI-based CAD in the pediatric radiology covering the brain [30,31,32,33,34,35,36,37,38], respiratory [42,43,44,45,46,47,48,49,50], musculoskeletal [40,41], urologic [51,52] and cardiac imaging [39]. Hence, it advances the previous two narrative reviews about various uses of AI in the pediatric radiology [17] and the AI-based CAD in the pediatric chest imaging [16] published in 2021 and 2022, respectively. Most of the included studies reported AI-based CAD model performances of at least 0.83 (AUC), 0.84 (sensitivity), 0.80 (specificity), 0.89 (PPV), 0.63 (NPV), 0.87 (accuracy), and 0.82 (F1 score) [30,31,32,33,34,35,36,37,38,39,40,41,42,43,44,45,46,47,48,49,50,51,52]. However, the diagnostic performances of these CAD systems appeared a bit lower than those reported in the systematic review of the AI-based CAD in the radiology (pooled sensitivity and specificity: 0.87 and 0.93, respectively) [10]. In addition, the pediatric pneumonia was the only disease that was investigated by more than two studies [43,45,46,47,48,49,50]. Although these studies reported that their CAD performances for the pneumonia diagnosis were at least 0.850 (AUC), 0.760 (sensitivity), 0.800 (specificity), 0.891 (PPV), 0.905 (accuracy) and 0.903 (F1 score), which would be sufficient to support less experienced pediatric radiologists in image interpretation, all but one were the retrospective studies and relied on the chest X-ray dataset consisting of 1741 normal and 4346 pneumonia images of 6087 1–5-year-old children collected from the Guangzhou Women and Children’s Medical Center, China [13,43,44,45,46,47,48,49,50]. It is noted that the use of the public dataset could facilitate AI-based CAD model performance comparison with other similar studies [43]. On the other hand, this approach would affect the model generalization ability (i.e., unable to maintain the performance when applying to different settings), causing the model to be unfit for real clinical situations [10,46]. Although techniques such as the cross-validation can be used to improve the AI-based CAD model generalization ability [37], only one of these studies used the cross-validation approach [50], while half of them did not report the internal validation type [43,45,49]. In addition, some ground truths given in the public datasets might be inaccurate, indicating potential reference standard issues [10,42]. These studies did not calculate the required sample size; perform the external validation; and compare their model performances with radiologists, but they are essential for the demonstration of the trustworthiness of study findings [43,45,46,48,49,50]. As per Table 2, the aforementioned methodological issues were also common for other included studies. These issues are found in many studies about the AI-based CAD in the radiology as well [10,12,13]. 

Table 2 reveals that the DL and its model, CNN, were commonly used for the development of the AI-based CAD systems in the pediatric radiology similar to the situation in the radiology [13]. According to the recent narrative review about the AI-based CAD in the pediatric chest imaging published in 2022, 144 Conformité Européenne-marked AI-based CAD systems for brain (35%), respiratory, (27%), musculoskeletal (11%), breast (11%), other (7%), abdominal (6%) and cardiac (4%) imaging were commercially available in the radiology [16]. The proportions of these systems are comparable to the findings of this systematic review that the brain, respiratory and musculoskeletal imaging were the three most popular application areas of the AI-based CAD in the pediatric radiology and the cardiac imaging was the least (Table 1). However, except for Helm et al.’s retrospective study about the detection of pediatric pulmonary nodules in 29 3–18-year-old patients with the use of the AI-based CAD system developed for adults [44], no commercial system was involved in the included studies (Table 2) [30,31,32,33,34,35,36,37,38,39,40,41,42,43,45,46,47,48,49,50,51,52]. Helm et al.’s study [44] was the only one that performed the external validation of the CAD system with the reference standard established by the consensus of six radiologists, and one of the few compared the CAD performance with the clinicians. However, that study only used four evaluation measures: sensitivity (0.42), specificity (1.00), PPV (1.00) and NPV (0.26), and the other metrics commonly used in more clinically focused studies, AUC and accuracy, were not reported [10,12,44,53]. This highlights that even for a more clinically focused AI-based CAD study in the pediatric radiology with the better design, the common methodological weaknesses such as the retrospective data collection with limited information of patient characteristics reported and cases included, and no sample size calculation, were still prevalent (Table 2) [44,54,55]. Hence, these explain the findings in Figure 2 that the concern regarding applicability was found in the patient selection, and the risk of bias was noted in both patient selection and reference standard categories, although similar results were also reported in the systematic reviews of the AI-based CAD in the radiology [10,12].

Apparently, the AI-based CAD in the pediatric radiology is less developed when compared to its adult counterpart. For example, not many studies were published before 2020 [33,35,36,44,52,56,57,58,59,60,61,62,63,64,65,66,67,68,69,70,71,72,73,74,75,76], and the studies mainly focused on the MRI and X-ray and particular patient cohorts [30,31,32,33,34,35,36,37,38,39,40,41,42,43,44,45,46,47,48,49,50,51,52] (Table 2). Although Schalekamp et al.’s [16] narrative review published in 2022 suggested the use of the AI-based CAD designed for the adult population in children, Helm et al.’s [44] study demonstrated that this approach yielded low sensitivity (0.42) and NPV (0.26) in detecting pediatric pulmonary nodules because of the smaller nodule sizes in children. Hence, AI-based CAD systems specifically designed/finetuned for the pediatric radiology by researchers and/or commercial companies seem necessary in the future. In addition, for further research, more robust study designs that can address the aforementioned methodological issues (especially the lack of the external validation) are essential for providing trustworthy findings to convince clinical centers to adopt the AI-based CAD in the pediatric radiology. In this way, the potential benefits of the CAD could be realized in a wider context [5,10,12,13]. 

This systematic review has two major limitations. The article selection, data extraction, and synthesis were performed by a single author, albeit one with more than 20 years of experience in conducting the literature reviews [14]. According to a recent methodological systematic review, this is an appropriate arrangement provided that the single reviewer is experienced [14,24,77,78,79]. Additionally, through adherence to the PRISMA guidelines and the use of the data extraction forms (Table 1 and Table 2) devised based on the recent systematic review on the diagnostic performance of the AI-based CAD in the radiology and the QUADAS-2 tool, the potential bias should be addressed to a certain extent [12,14,26,29]. In addition, only articles in English identified via databases were included, potentially affecting the comprehensiveness of this systematic review [9,21,26,27,80]. Nevertheless, this review still has a wider coverage about the AI-based CAD in the pediatric radiology than the previous two narrative reviews [16,17].

## 5. Conclusions

This systematic review shows that the AI-based CAD for the pediatric radiology could be applied in the brain, respiratory, musculoskeletal, urologic and cardiac imaging. Most of the studies (93.3%, 14/15; 77.8%, 14/18; 73.3%, 11/15; 80.0%, 8/10; 66.6%, 2/3; 84.2%, 16/19; 80.0%, 8/10) reported AI-based CAD model performances of at least 0.83 (AUC), 0.84 (sensitivity), 0.80 (specificity), 0.89 (PPV), 0.63 (NPV), 0.87 (accuracy), and 0.82 (F1 score), respectively. The pediatric pneumonia was the most common pathology covered in the included studies. They reported that their CAD performances for pneumonia diagnosis were at least 0.850 (AUC), 0.760 (sensitivity), 0.800 (specificity), 0.891 (PPV), 0.905 (accuracy) and 0.903 (F1 score). Although these diagnostic performances appear sufficient to support the less experienced pediatric radiologists in the image interpretation, a range of methodological weaknesses such as the retrospective data collection, no sample size calculation, overreliance on public dataset, small test set size, limited patient cohort coverage, use of diagnostic accuracy measures and cross-validation, lack of model external validation and model performance comparison with clinicians, and risk of bias of reference standard are found in the included studies. Hence, their AI-based CAD systems might be unfit for the real clinical situations due to a lack of generalization ability. In the future, more AI-based CAD systems specifically designed/fine-tuned for a wider range of imaging modalities and pathologies in the pediatric radiology should be developed. In addition, more robust study designs should be used in further research to address the aforementioned methodological issues for providing the trustworthy findings to convince the clinical centers to adopt the AI-based CAD in the pediatric radiology. In this way, the potential benefits of the CAD could be realized in a wider context.

## Figures and Tables

**Figure 1 children-10-00525-f001:**
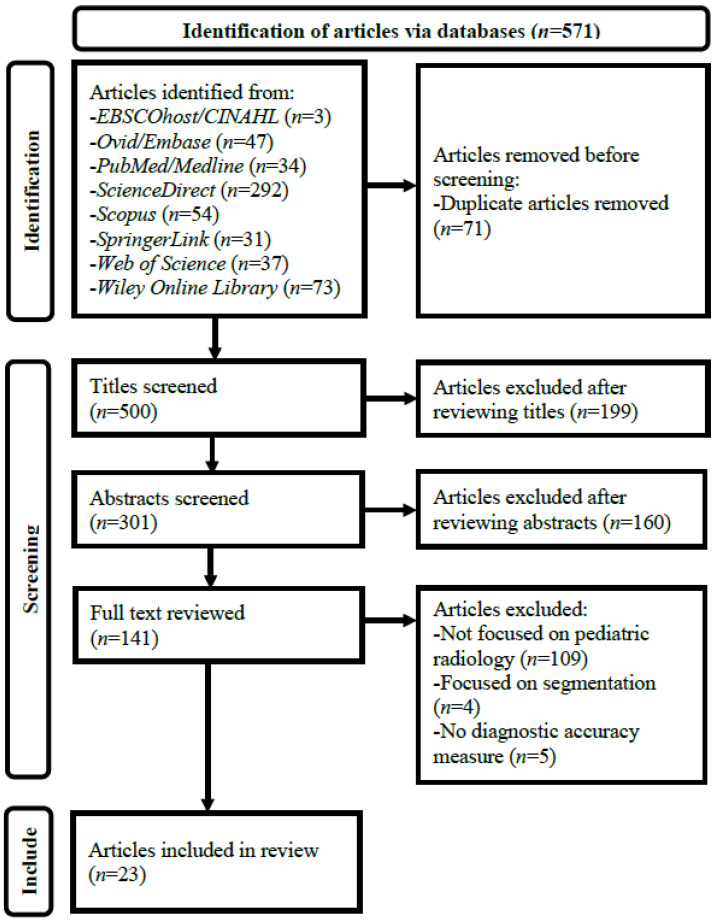
Preferred reporting items for systematic reviews and meta-analyses flow diagram for systematic review of diagnostic performance of artificial intelligence-based computer-aided detection and diagnosis in pediatric radiology. CINAHL, Cumulative Index of Nursing and Allied Health Literature.

**Figure 2 children-10-00525-f002:**
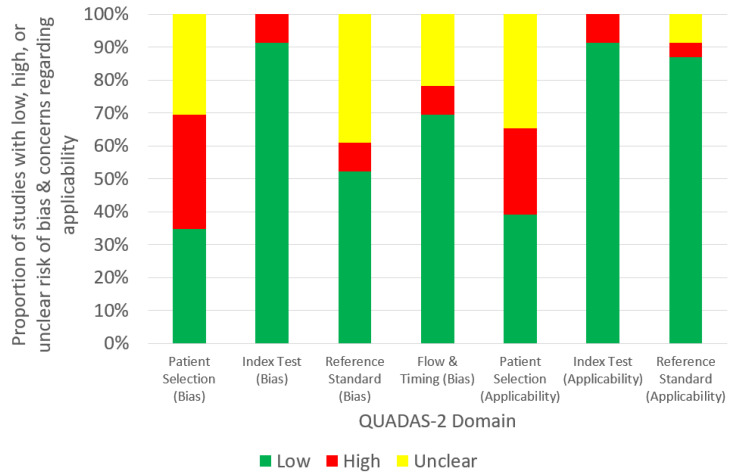
Quality assessment summary of all (23) included studies based on Revised Quality Assessment of Diagnostic Accuracy Studies tool.

## Data Availability

Not applicable.

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
