# Peer review of "Diagnostic Performance of Artificial Intelligence-Based Computer-Aided Detection and Diagnosis in Pediatric Radiology: A Systematic Review"

_children, 2023, doi:10.3390/children10030525_

Round 1

Reviewer 1 Report

The manuscript can be accepted after the authors correct the following comments:

Comments per Section of Manuscript:

Title

1.      The title clearly and sufficiently reflects its content.

Abstract

  1. Please note that the main aims of abstract section should not be as a question. Please rephrase it as a sentence.
  2. In the abstract section or in any section in whole manuscript, when you mention the word for the first time, please define it as words and not as an abbreviation.
  3. The abstract should be rewrite.

Introduction

  1. Please note that the introduction part in the review article needs at least four comprehensive paragraphs. So, kindly write further paragraphs in the introduction section.
  2. Increase number of references in the introduction part. This is a review article and requires several references (more than 100 references).
  3. The academic writing is written well.

Methodology

1.      The review article has no method section.  So, I suggest to start directly with section  2.1.

Results

1.      The result section is written well.

Discussion and Conclusion

No comments.

References

  1. This is a review article and requires several references. At least 100 references.

Author Response

Title

  1. The title clearly and sufficiently reflects its content.

Response: Thank you for your comment.

Abstract

  1. Please note that the main aims of abstract section should not be as a question. Please rephrase it as a sentence.

Response: Thank you for your comment. The purpose has been changed from “The purpose of this systematic review is to answer the question “What are the AI-based CAD ap-plications in pediatric radiology, their diagnostic performances and methods for their performance evaluation?”” to “The purpose of this systematic review is to investigate the AI-based CAD applications in pediatric radiology, their diagnostic performances and methods for their performance evaluation.” to address this comment.

  1. In the abstract section or in any section in whole manuscript, when you mention the word for the first time, please define it as words and not as an abbreviation.

Response: Thank you for your comment. The use of abbreviations in my paper has been reviewed. The following abbreviations have been defined when appearing in the first time.

  • Lines 71-72: preferred reporting items for systematic reviews and meta-analyses (PRISMA)
  • Lines 108-109: CINAHL, Cumulative Index of Nursing and Allied Health Literature
  • Line 156: UK, United Kingdom
  • Lines 156-157: USA, United States of America
  • Line 191: ML, machine learning; MRI, magnetic resonance imaging
  • Line 192: UK, United Kingdom; USA, United States of America

The abbreviations below have been removed.

  • Line 39: USA
  • Line 40: FDA
  • Line 73: PICO
  • Line 77: CINAHL
  • Line 106: PRISMA
  • Line 117: SVM
  • Line 190: LSTM
  • Line 203: QUADAS-2
  • Line 241: CE

I hope these changes have addressed your comment.

  1. The abstract should be rewritten.

Response: Thank you for your comment. The abstract has been rewritten as per your previous suggestion for addressing this comment.

Introduction

  1. Please note that the introduction part in the review article needs at least four comprehensive paragraphs. So, kindly write further paragraphs in the introduction section.

Response: Thank you for your comment. The number of paragraphs in the Introduction section has been changed from 3 to 4. Also, I have checked my paper’s references, the recent systematic reviews about artificial intelligence-based computer aided detection and diagnosis in radiology (refs 10 and 12) that were published in 2019 and 2021 in the journals, The Lancet Digital Health (impact factor [IF]: 36.615) and npj Digital Medicine (IF: 15.357) with IFs higher than the Children journal (IF: 2.835), and the numbers of words within their Introduction sections were 377 and 357 respectively. For my revised Introduction section, it has 437 words. Hence, it should be comprehensive enough. I hope the change has addressed your comment.

  1. Increase number of references in the introduction part. This is a review article and requires several references (more than 100 references).

Response: Thank you for your comment. I have checked my paper’s references 10 and 12 stated above and the numbers of references cited in their Introduction parts were 16 and 13 respectively. The number of references cited in the Introduction section of my revised paper is 17 which is greater than those published in the high impact journals, Lancet Digital Health (IF: 36.615) and npj Digital Medicine (IF: 15.357). Also, I have checked other references of my paper, the recent systematic reviews about artificial intelligence-based computer aided detection and diagnosis in radiology (refs 21 and 27) that were published in the journals, European Journal of Radiology (IF: 4.531) and European Thyroid Journal (IF: 4.084) in 2022 and 2020 with IFs higher than the Children journal (IF: 2.835), and the total numbers of references cited in those systematic reviews were 45 and 33 respectively. For my revised paper, it has 53 references in total. Hence, the numbers of references included in my Introduction section and the whole review should be enough and able to meet the current publication standards. I hope you will find my response satisfactory.

  1. The academic writing is written well.

Response: Thank you for your comment.

Methodology

  1. The review article has no method section. So, I suggest to start directly with section 2.1.

Response: Thank you for your comment. I have checked several latest systematic reviews published in the Children journal and their details are as follows.

  • Azevedo, A.C.; Hilário, S.; Gonçalves, M.F.M. Microbiome in Nasal Mucosa of Children and Adolescents with Allergic Rhinitis: A Systematic Review. Children 2023, 10, 226. https://doi.org/10.3390/children10020226
  • Mohammed, I.E.; Shariff, N.; Mohd Hanim, M.F.; Mohd Yusof, M.Y.P.; Md Sabri, B.A.; Md Bohari, N.F.; Venkiteswaran, A. Knowledge, Attitudes and Professional Behavior of Silver Diamine Fluoride among Dental Personnel: A Systematic Review. Children 2022, 9, 1936. https://doi.org/10.3390/children9121936
  • Banzato, A.; Cerchiari, A.; Pezzola, S.; Ranucci, M.; Scarfò, E.; Berardi, A.; Tofani, M.; Galeoto, G. Evaluation of the Effectiveness of Functional Chewing Training Compared with Standard Treatment in a Population of Children with Cerebral Palsy: A Systematic Review of Randomized Controlled Trials. Children 2022, 9, 1876. https://doi.org/10.3390/children9121876

These review articles reported the guidelines used in Section 2 Materials and Methods before Section 2.1. For my paper, I also reported the guidelines used in Section 2 Materials and Methods before Section 2.1. Hence, I believe my writing style is in line with the publication standards of the Children journal. I hope you will find my response satisfactory.

Results

  1. The result section is written well.

Response: Thank you for your comment.

Discussion and Conclusion

  1. No comments.

Response: Noted.

References

  1. This is a review article and requires several references. At least 100 references.

Response: Thank you for your comment. I have checked my paper’s references, the recent systematic reviews about artificial intelligence-based computer aided detection and diagnosis in radiology (refs 21 and 27) that were published in the journals, European Journal of Radiology (IF: 4.531) and European Thyroid Journal (IF: 4.084) in 2022 and 2020 with IFs higher than the Children journal (IF: 2.835), and the total numbers of references cited in those systematic reviews were 45 and 33 respectively. Also, for the 3 aforementioned systematic reviews published in the Children journal, their total numbers of included references were between 42 and 47. For my revised paper, it has 53 references in total. Hence, the total number of references included in my revised review should be enough and able to meet the current publication standards. I hope you will find my response satisfactory.

Reviewer 2 Report

The manuscript entitled “Diagnostic Performance of Artificial Intelligence-Based Computer-Aided Detection and Diagnosis in Pediatric Radiology: A Systematic Review “ has been investigated in detail.
The author also provided a comprehensive overview of previous works in this field, highlighting the strengths and limitations of each approach. The paper is well-written and easy to follow, making it accessible to a wide audience.

The paper’s subject could be interesting for readers of journal. Therefore, I recommend this paper for publication in this journal but before that, I have a few comments on the text that should be addressed before publication:

Comments:

1) The author mentioned these numbers in the abstract “Most of the included studies reported their model performances of at least 0.83 (area under receiver operating characteristic curve), 0.8 (sensitivity), 0.80 (specificity), 0.89 (positive predictive value), 0.63 (negative predictive value), 0.87 (accuracy), and 0.82 (F1 score).” What do they mean exactly?  

2) There are lots of grammatical mistakes in the manuscript. Please correct them.

3) Some abbreviations haven’t been described for the first time in the text. Please explain all in whole of the manuscript.

4) in the conclusion section, the numbers used to imply the performance accuracy of previous works, must be explained. If they’re error. It should be explained which type of error is that.

5)It would be nice if some new references (such as following ref) in field of application of AI in medical imaging systems are added to the text.

[1]Lu, Y.; Zheng, N.; Ye, M.; Zhu, Y.. Proposing Intelligent Approach to Predicting Air Kerma within Radiation Beams of Medical X-ray Imaging Systems. Diagnostics 2023, 13, 190. https://doi.org/10.3390/diagnostics13020190

6)The paper is well organized and I’m pretty happy with that

Author Response

The manuscript entitled “Diagnostic Performance of Artificial Intelligence-Based Computer-Aided Detection and Diagnosis in Pediatric Radiology: A Systematic Review “ has been investigated in detail.

The author also provided a comprehensive overview of previous works in this field, highlighting the strengths and limitations of each approach. The paper is well-written and easy to follow, making it accessible to a wide audience.

The paper’s subject could be interesting for readers of journal. Therefore, I recommend this paper for publication in this journal but before that, I have a few comments on the text that should be addressed before publication:

Response: Thank you for your comment.

Comments:

1) The author mentioned these numbers in the abstract “Most of the included studies reported their model performances of at least 0.83 (area under receiver operating characteristic curve), 0.8 (sensitivity), 0.80 (specificity), 0.89 (positive predictive value), 0.63 (negative predictive value), 0.87 (accuracy), and 0.82 (F1 score).” What do they mean exactly? 

Response: Thank you for your comment. The sentence, “Most of the included studies reported their model performances of at least 0.83 (area under receiver operating characteristic curve), 0.84 (sensitivity), 0.80 (specificity), 0.89 (positive predictive value), 0.63 (negative predictive value), 0.87 (accuracy), and 0.82 (F1 score).” in the abstract has been changed to “Most of the studies (93.3%, 14/15; 77.8%, 14/18; 73.3%, 11/15; 80.0%, 8/10; 66.6%, 2/3; 84.2%, 16/19; 80.0%, 8/10) reported their model performances of at least 0.83 (area under receiver operating characteristic curve), 0.84 (sensitivity), 0.80 (specificity), 0.89 (positive predictive value), 0.63 (negative predictive value), 0.87 (accuracy), and 0.82 (F1 score), respectively. ” for clarifying the point and addressing this comment.

2) There are lots of grammatical mistakes in the manuscript. Please correct them.

Response: Thank you for your comment. The manuscript has been reviewed. The grammatical mistakes have been corrected for addressing this comment.

3) Some abbreviations haven’t been described for the first time in the text. Please explain all in whole of the manuscript.

Response: Thank you for your comment. The use of abbreviations in my paper has been reviewed. The following abbreviations have been defined when appearing in the first time.

  • Lines 71-72: preferred reporting items for systematic reviews and meta-analyses (PRISMA)
  • Lines 108-109: CINAHL, Cumulative Index of Nursing and Allied Health Literature
  • Line 156: UK, United Kingdom
  • Lines 156-157: USA, United States of America
  • Line 191: ML, machine learning; MRI, magnetic resonance imaging
  • Line 192: UK, United Kingdom; USA, United States of America

The abbreviations below have been removed.

  • Line 39: USA
  • Line 40: FDA
  • Line 73: PICO
  • Line 77: CINAHL
  • Line 106: PRISMA
  • Line 117: SVM
  • Line 190: LSTM
  • Line 203: QUADAS-2
  • Line 241: CE

I hope these changes have addressed your comment.

4) in the conclusion section, the numbers used to imply the performance accuracy of previous works, must be explained. If they’re error. It should be explained which type of error is that.

Response: Thank you for your comment. The sentence, “Most of the included studies reported their AI-based CAD model performances of at least 0.83 (AUC), 0.84 (sensitivity), 0.80 (specificity), 0.89 (PPV), 0.63 (NPV), 0.87 (ac-curacy), and 0.82 (F1 score).” in the conclusions section has been changed to “Most of the studies (93.3%, 14/15; 77.8%, 14/18; 73.3%, 11/15; 80.0%, 8/10; 66.6%, 2/3; 84.2%, 16/19; 80.0%, 8/10) reported their model performances of at least 0.83 (area under receiver operating characteristic curve), 0.84 (sensitivity), 0.80 (specificity), 0.89 (positive predictive value), 0.63 (negative predictive value), 0.87 (accuracy), and 0.82 (F1 score), respectively.” for clarifying the point and addressing this comment.

5)It would be nice if some new references (such as following ref) in field of application of AI in medical imaging systems are added to the text.

[1]Lu, Y.; Zheng, N.; Ye, M.; Zhu, Y.. Proposing Intelligent Approach to Predicting Air Kerma within Radiation Beams of Medical X-ray Imaging Systems. Diagnostics 2023, 13, 190. https://doi.org/10.3390/diagnostics13020190

Response: Thank you for your comment. The suggested reference has been added for addressing this comment.

6)The paper is well organized and I’m pretty happy with that

Response: Thank you for your comment.

Round 2

Reviewer 2 Report

all comments have been addressed correctly